# 3DRP-Net: 3D Relative Position-aware Network for 3D Visual Grounding

**Zehan Wang[1]*   Haifeng Huang[1]*   Yang Zhao[2]   Linjun Li[1]   Xize Cheng[1]**
**Yichen Zhu[1]   Aoxiong Yin[1]   Zhou Zhao[1]†**
[1]Zhejiang University   [2]ByteDance
*{wangzehan01, huanghaifeng}@zju.edu.cn*

## Abstract

3D visual grounding aims to localize the target object in a 3D point cloud by a free-form language description. Typically, the sentences describing the target object tend to provide information about its relative relation between other objects and its position within the whole scene. In this work, we propose a relation-aware one-stage framework, named **3D R**elative **P**osition-aware **Net**work (3DRP-Net), which can effectively capture the relative spatial relationships between objects and enhance object attributes. Specifically, 1) we propose a **3D R**elative **P**osition **M**ulti-head **A**ttention (3DRP-MA) module to analyze relative relations from different directions in the context of object pairs, which helps the model to focus on the specific object relations mentioned in the sentence. 2) We designed a soft-labeling strategy to alleviate the spatial ambiguity caused by redundant points, which further stabilizes and enhances the learning process through a constant and discriminative distribution. Extensive experiments conducted on three benchmarks (i.e., ScanRefer and Nr3D/Sr3D) demonstrate that our method outperforms all the state-of-the-art methods in general.

## 1 Introduction

Visual grounding aims to localize the desired objects based on the given natural language description. With the rapid development and wide applications of 3D vision (Xia et al., 2018; Savva et al., 2019; Zhu et al., 2020; Wang et al., 2019) in recent years, 3D visual grounding task has received more and more attention. Compared to the well-studied 2D visual grounding (Yang et al., 2019; Kamath et al., 2021; Yang et al., 2022; Li and Sigal, 2021; Deng et al., 2021; Plummer et al., 2015; Kazemzadeh et al., 2014), the input sparse point clouds in the 3D visual grounding task are more

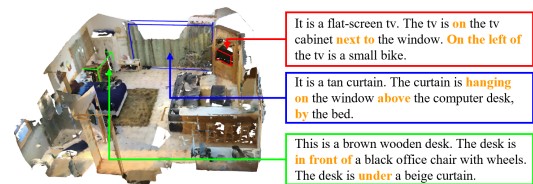

Figure 1: 3D visual grounding is the task of grounding a description in a 3D scene. In the sentences, all the words indicating the relative positions of the target object are **bolded**. Notice that relative position relations between objects are crucial for distinguishing the target object, and the relative position-related descriptions in 3D space are complex (e.g., "above", "on the left", "in front of", and "next to", etc.)

irregular and more complex in terms of spatial positional relationships, which makes it much more challenging to locate the target object.

In the field of 3D visual grounding, the previous methods can be mainly categorized into two groups: the two-stage approaches (Chen et al., 2020; Achlioptas et al., 2020; Zhao et al., 2021b; Yuan et al., 2021; Huang et al., 2022; Cai et al., 2022; Huang et al., 2021; Wang et al., 2023) and the one-stage approaches (Luo et al., 2022). The former ones follow the detection-and-rank paradigm, and thanks to the flexibility of this architecture, they mainly explore the benefits of different object relation modeling methods for discriminating the target object. The latter fuse visual-text features to predict the bounding boxes of the target objects directly, and enhance the object attribute representation by removing the unreliable proposal generation phase.

However, these two methods still have limitations. For two-stage methods, the model performance is highly dependent on the quality of the object proposals. However, due to the sparsity and irregularity of the input 3D point cloud, sparse proposals may leave out the target object, while dense proposals will bring redundant computational costs and make the matching stage too

---

*Equal contribution.
†Corresponding author.

complicated to distinguish the target object. As for the one-stage methods, although the existing approach (Luo et al., 2022) achieves better performance, they can not capture the relative spatial relationships between objects, which makes it often fail in samples that rely on relative relation reasoning. As shown in Fig.1, the majority of sentences in 3D visual grounding contain relative spatial relation descriptions. Furthermore, due to the spatial complexity of the 3D scene, there are various relative position-related descriptions from different orientations. To further illustrate that relative position is a general and fundamental issue in 3D visual grounding tasks, we analyze the frequency of relative position words in ScanRefer and Nr3D/Sr3D, and the results show that at least $90\%$ of the sentences describe the relative position of objects, and most of them contain multiple spatial relations. Detailed statistics can be found in supplementary materials.

To alleviate above problems, we propose a one-stage 3D visual grounding framework, named **3D R**elative **P**osition-aware **Net**work (3DRP-Net). Our 3DRP-Net combines and enhances the advantages of the two-stage approaches for relations modeling and the one-stage approaches for proposal-free detection while avoiding the shortcomings of both methods. For the relations modeling, we devise a novel **3D R**elative **P**osition **M**ulti-head **A**ttention (3DRP-MA) module, which can capture object relations along multiple directions and fully consider the interaction between the relative position and object pairs which is ignored in previous two-stage methods (Yuan et al., 2021; Zhao et al., 2021b; Huang et al., 2021).

Specifically, we first extract features from the point cloud and description, and select key points. Then, the language and visual features interacted while considering the relative relations between objects. For the relation modeling, We introduce learnable relative position encoding in different heads of the multi-head attention to capture object pair relations from different orientations. Moreover, in sentences, the relative relations between objects are usually described as *"Object 1-Relation-Object 2"*, such as "tv is on the tv cabinet" and "curtain is hanging on the window" in Fig.1. The relation is meaningful only in the context of object pairs, thus our relative position encoding would interact with the object pairs' feature, to better capture and focus on the mentioned relations.

Besides, as discussed in (Qi et al., 2019), point clouds only capture surface of object, and the 3D object centers are likely to be far away from any point. To accurately reflect the location of objects and learn comprehensive object relation knowledge, we sample multiple key points of each object. However, redundant key points may lead to ambiguity. To achieve disambiguation while promoting a more stable and discriminative learning process, we propose a soft-labeling strategy that uses a constant and discriminative distribution as the target label instead of relying on unstable and polarized hard-label or IoU scores.

Our main contributions can be summarized as follows:

- We propose a novel single-stage 3D visual grounding model, called **3D R**elative **P**osition-aware **Net**work (3DRP-Net), which for the first time captures relative position relationships in the context of object pairs for better spatial relation reasoning.

- We design a **3D R**elative **P**osition **M**ulti-head **A**ttention (3DRP-MA) module for simultaneously modeling spatial relations from different orientations of 3D space. Besides, we devise a soft-labeling strategy to alleviate the ambiguity while further enhancing the discriminative ability of the optimal key point and stabilizing the learning process.

- Extensive experiments demonstrate the effectiveness of our method. Our 3DRP-Net achieves state-of-the-art performance on three mainstream benchmark datasets ScanRefer, Nr3D, and Sr3D.

## 2 Related Work

### 2.1 3D Visual Grounding

Recent works in 3D visual grounding can be summarized in two categories: two-stage and one-stage methods. We briefly review them in the following. **Two-stage Methods.** Two-stage approaches follow the detection-and-rank scheme. In the first stage, 3D object proposals are generated by a pre-trained 3D object detector (Chen et al., 2020) or with the ground truth (Achlioptas et al., 2020). In the second stage, the best matching proposals would be selected by leveraging the language description. Advanced two-stage methods achieve good performance by better modeling the relationships among objects. Referit3D (Achlioptas et al.,

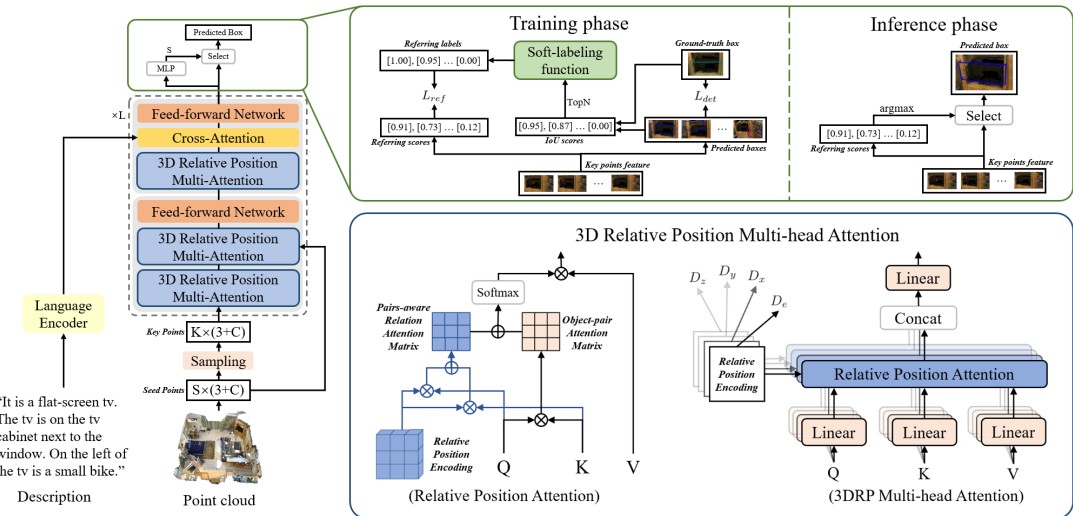

Figure 2: 3DRP-Net is a transformer-based one-stage 3D VG model which takes a 3D point cloud and a description as inputs and outputs the bounding box of the object most relevant to the input expression. In the stacked transformer layer, the 3DRP-MA captures the relative relations between points in the 3D perspective. Specifically, the two self-attentions based on 3DRP-MA capture the relative relations between objects, while the cross-attention between key points and seed points enhances the global position information.

2020) and TGNN (Huang et al., 2021) make use of the graph neural network (Scarselli et al., 2008) to model the relationships between objects. 3DVG-Transformer (Zhao et al., 2021b) utilize attention mechanisms (Vaswani et al., 2017) to enable interactions between proposals, and the similarity matrix can be adjusted based on the relative Euclidean distances between each pair of proposals.

**One-stage Methods.** One-stage approaches avoid the unstable and time-consuming object proposals generation stage under the detection-and-rank paradigm. The visual features extracted by the backbone are directly and densely fused with the language features, and the fused features are leveraged to predict the bounding boxes and referring scores. 3D-SPS (Luo et al., 2022) first addresses the 3D visual grounding problem by one-stage strategy. It firstly filters out the key points of language-relevant objects and processes inter-model interaction to progressively down-sample the key points.

Our work utilizes the advanced one-stage framework and introduces a novel relative relation module to effectively capture the intricate relations between objects, enabling our model to achieve superior performance.

## 2.2 Position Encoding in Attention

The attention mechanism is the primary component of transformer (Vaswani et al., 2017). Since the attention mechanism is order-independent, infor-

mation about the position should be injected for each token. In general, there are two mainstream encoding methods: absolute and relative position encoding.

**Absolute Position Encoding.** The original transformer (Vaswani et al., 2017) considers the absolute positions, and the encodings are generated based on the sinusoids of varying frequency. Recent 3D object detection studies also use absolute position encodings. In Group-free (Liu et al., 2021b), the encodings are learned by the center and size of the predicted bounding box, while the Fourier function is used in 3DETR (Misra et al., 2021).

**Relative Position Encoding.** Recently, some advanced works in natural language processing (He et al., 2020; Raffel et al., 2020; Shaw et al., 2018) and image understanding (Liu et al., 2021a; Hu et al., 2019, 2018) generate position encoding based on the relative distance between tokens. Relative relation representations are important for tasks where the relative ordering or distance matters.

Our method extends relative position encoding to 3D Euclidean space and enhances relative relation reasoning ability in 3D visual grounding.

## 3 Method

This section introduces the proposed 3D Relative Position-aware Network (3DRP-Net) for 3D visual grounding. In Sec.3.1, we present an overview of our method. In Sec.3.2, we dive into the techni-

cal details of the 3D Relative Position Multi-head Attention (3DRP-MA) module and how to comprehensively and efficiently exploit the spatial position relations in the context of object pairs. In Sec.3.3 and Sec.3.4, we introduce our soft-labeling strategy and the training objective function of our method.

## 3.1 Overview

The 3D visual grounding task aims to find the object most relevant to a given textual query. So there are two inputs in the 3D visual grounding task. One is the 3D point cloud which is represented by the 3D coordinates and auxiliary features (RGB values and normal vectors in our setting) of $N$ points. Another input is a free-form natural language description with $L$ words.

The overall architecture of our 3DRP-Net is illustrated in Fig.2. *Firstly*, we adopt the pre-trained PointNet++ (Qi et al., 2017) to sample $S$ seed points and $K$ key points from the input 3d point cloud and extract the $C$-dimensional enriched points feature. For the language description input, by using a pre-trained language encoder (Radford et al., 2021), we encode the $L$-length sentences to $D$-dimensional word features. *Secondly*, a stack of transformer layers are applied for multimodal fusion. The features of key points are accordingly interacted with language and seed points to group the scene and language information for detection and localization. Our new 3D relative position multi-head attention in each layer enables the model to understand vital relative relations among objects in the context of each object pair. *Eventually*, we use two standard multi-layer perceptrons to regress the bounding box and predict the referring confidence score based on the feature of each key point. As shown in Fig.2, in the training phase, we generate the target labels of referring scores based on the IoUs of the predicted boxes. During inference, we only select the key point with the highest referring score to regress the target bounding box.

## 3.2 3D Relative Position Multi-head Attention

When describing an object in 3D space, relations between objects are essential to distinguish objects in the same class. Given the spatial complexity of 3D space and the potentially misleading similar relative positions between different object pairs, a precise and thorough comprehension of the relative position relationships is crucial for 3D visual grounding. However, existing 3D visual grounding methods fail to effectively address complex spa-

tial reasoning challenges, thereby compromising their performance. To address this limitation, we propose a novel 3D relative position multi-head attention to model object relations in the context of corresponding object pairs within an advanced one-stage framework.

### 3.2.1 Relative Position Attention

Before detailing our relative position attention, we briefly review the original attention mechanism in (Vaswani et al., 2017). Given an input sequence $x = \{x_1, ..., x_n\}$ of $n$ elements where $x_i \in \mathbb{R}^{d_x}$, and the output sequence $z = \{z_1, ..., z_n\}$ with the same length where $z_i \in \mathbb{R}^{d_z}$. Taking single-head attention, the output can be formulated as:

$$q_i = x_i W^Q, \; k_j = x_j W^K, \; v_i = x_i W^V \quad (1)$$

$$a_{i,j} = \frac{q_i k_j^T}{\sqrt{d}}, \; z_i = \sum_{j=1}^{n} \frac{exp(a_{i,j})}{\sum_{k=1}^{n} exp(a_{i,k})} v_j \quad (2)$$

where $W_Q, W_K, W_V \in \mathbb{R}^{d_x \times d_z}$ represents the projection matrices, $a_{i,j}$ is the attention weight from element $i$ to $j$.

Based on the original attention mechanism, we propose a novel relative position attention that incorporates relative position encoding between elements. Since the semantic meaning of a relative relation *"Object 1-Relation-Object 2"* is also highly dependent on the object pairs involved, it is essential for the position encoding to fully interact with object features in order to accurately capture the specific relative relations mentioned in the description. To this end, the attention weight $a_{i,j}$ in our proposed relative position attention is calculated as follows:

$$a_{i,j} = \frac{q_i k_j^T + q_i {r_{p(d_{ij})}^k}^T + r_{p(d_{ji})}^q k_j^T}{\sqrt{3d}} \quad (3)$$

where $d_{ij}$ represents the relative distance from element $i$ to element $j$, while $d_{ji}$ is the opposite. $p(d) \in [0, 2k)$ is an index function that maps continuous distance to discrete value, as detailed in Eq.4. $r_{p(\cdot)}^k, r_{p(\cdot)}^q \in \mathbb{R}^{(2k+1) \times d_z}$ is the learnable relative position encoding. Considering a typical object relation expression *"Object 1-Relation-Object 2"*, our attention weight can be understood as a sum of three attention scores on object pairs and relation: *Object 1-to-Object 2*, *Object 1-to-Relation*, and *Relation-to-Object 2*.

### 3.2.2 Piecewise Index Function

The points in the 3D point cloud are unevenly distributed in a Euclidean space, and the relative distances are continuous. To enhance the relative spatial information and reduce computation costs, we propose to map the continuous 3D relative distances into discrete integers in a finite set. Inspired by (Wu et al., 2021), we use the following piecewise index function:

$$p(d) = \begin{cases} [d], & |d| \leqq \alpha \\ sign(d) \times min(k, [\alpha + \frac{ln(|d|/\alpha)}{ln(\beta/\alpha)}(k - \alpha)]), & |d| > \alpha \end{cases}$$
(4)

where $[\cdot]$ is a round operation, $sign(\cdot)$ represents the sign of a number, i.e., returning 1 for positive input, -1 for negative, and 0 for otherwise.

Eq.4 performs a fine mapping in the $\alpha$ range. The further over $\alpha$, the coarser it is, and distances beyond $\beta$ would be mapped to the same value. In the 3D understanding field, many studies (Zhao et al., 2021a; Misra et al., 2021) have demonstrated that neighboring points are much more important than the further ones. Therefore, mapping from continuous space to discrete values by Eq.4 would not lead to much semantic information loss while significantly reducing computational costs.

### 3.2.3 Multi-head Attention for 3D Position

Till now, our relative position attention module can handle the interaction between object features and relative position information in continuous space. However, points in 3D space have much more complicated spatial relations than pixels in 2D images or words in 1D sentences. As shown in Table 4, relying on a single relative distance metric leads to insufficient and partial capture of inter-object relations. This makes it difficult to distinguish the target object when multiple spatial relations are described in the language expression. Therefore, we attempt to capture object relations from multiple directions. Specifically, we encode the relative distances under x, y, z coordinates, and the Euclidean metric, denoted as $D_x$, $D_y$, $D_z$, and $D_e$, respectively. These four relative position metrics represent most of object relations in the language description (e.g., $D_x$ for "left, right", $D_y$ for "front, behind", $D_z$ for "top, bottom", $D_e$ for "near, far"). Based on the architecture of multi-head attention, each relative position encoding is injected into the relative position attention module of each head. Such a 3DRP-MA allows the model to jointly attend to information from different relative relations in 3D space.

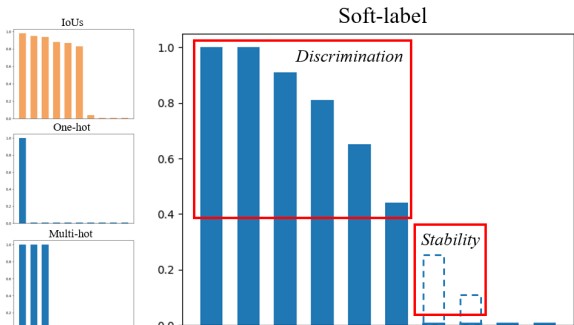

Figure 3: Comparison of various labeling strategies.

### 3.3 Soft-labeling Strategy

Due to the object center are often not contained in the given point clouds, we select multiple key points for each object to better reflect its location. Therefore, as shown in Fig.3, there will be lots of accurately predicted boxes achieving high Intersection over Union (IoU) of target object. Previous methods (Chen et al., 2020; Zhao et al., 2021b; Luo et al., 2022) use one-hot or multi-hot labels to supervise the referring score. The key points whose predicted box has the top $N_s$ highest IoU are set to 1, and others are set to 0, which can encourage the model to select the most high-IoU proposals. However, the simple hard-labeling strategy results in two problems: Firstly, proposals with similar and high IoUs may be labeled differently as 1 and 0, which can cause an unstable training phase. Secondly, it becomes difficult to distinguish between optimal and sub-optimal proposals, affecting the model's ability to accurately identify the most accurate proposal.

To tackle these issues, we introduce a soft-labeling strategy to smooth the label distribution and encourage the model to effectively distinguish the optimal proposal. To be specific, the soft-labeling function can be calculated as follow:

$$\hat{s}_i = exp(-\frac{i^2}{2\sigma^2} + 1)$$
(5)

where $i \in \{0, ..., N_s\}$ represents the $i$-th highest IoU. We set $\sigma$ as $[N_s/3]$ to control the smoothness of the distributions. The target label of the keypoint whose predicted box's IoU is $i$-th highest and greater than 0.25 is set to $\hat{s}_i$, and others are set to 0.

Although this strategy is simple, its role is to do more as one stroke, and the insight it provides is non-trivial.

*For discriminative ability*, the soft-labels enhance the difference between the optimal and sub-

optimal proposals, which enforces the model to accurately identify the best key point for regressing detection box. In contrast, when hard-labels or IoU scores are used as the target labels, there is little difference between optimal and sub-optimal proposals from the perspective of learning objectives. *For stability*, compared to hard-labels, our soft-labels can cover a broader range of accurate proposals with a smoother label distribution, and excluding the proposals with low IoU further stabilizes the learning process. Additionally, compared to directly using IoU scores, the constant distribution in soft-labels provides a more stable loss across different samples. For example, if we have two samples with vastly different target objects, such as a large bed and a small chair, the bed sample would have significantly more key points selected, resulting in more proposals of the target object. Using IoU scores as labels would ultimately lead to a much larger loss for the bed sample than the chair sample, which is clearly unreasonable.

## 3.4 Training and Inference

We apply a multi-task loss function to train our 3DRP-Net in an end-to-end manner.

**Referring Loss.** The Referring loss $L_{ref}$ is calculated between the target labels $\hat{S}$ discussed in Sec.3.3 and predicted referring scores $S$ of $K$ key-points with focal loss (Lin et al., 2017).

**Keypoints Sampling Loss.** Following the loss used in (Luo et al., 2022), we apply the key points sampling loss $L_{ks}$ to make sure the selected key points are relevant to any object whose category is mentioned in the description.

**Detection Loss.** To supervise the predicted bounding boxes, we use the detection loss $L_{det}$ as an auxiliary loss. Following (Luo et al., 2022), the $L_{det}$ consists of semantic classification loss, objectness binary classification loss, center offset regression loss and bounding box regression loss.

**Language Classification Loss.** Similar to (Chen et al., 2020), We introduce the language classification loss $L_{text}$ to enhance language encoder.

Finally, the overall loss function in the training process can be summarized as

$$L = \alpha_1 L_{ref} + \alpha_2 L_{ks} + \alpha_3 L_{det} + \alpha_4 L_{text} \quad (6)$$

where the balancing factors $\alpha_1$, $\alpha_2$, $\alpha_3$, $\alpha_4$ are set default as 0.05, 0.8, 5, 0.1, respectively, and the $L_{ref}$ and $L_{det}$ are applied on all decoder stages following the setting in (Qi et al., 2019).

## 4 Experiment

### 4.1 Datasets and Metrics

**ScanRefer.** The ScanRefer dataset (Chen et al., 2020) annotates 800 scenes with 51,583 language descriptions based on ScanNet dataset (Dai et al., 2017). Following the ScanRefer benchmark, we split the train/val/test set with 36,655, 9,508, and 5,410 samples, respectively.

**Nr3D/Sr3D.** The Nr3D and Sr3D are two subdatasets in ReferIt3D (Achlioptas et al., 2020). They are also annotated on the indoor 3D scene dataset Scannet (Dai et al., 2017). Nr3D contains 41,503 human utterances collected by ReferItGame, and Sr3D contains 83,572 synthetic descriptions generated based on a "target-spatial relationship-anchor object" template.

**Evaluation Metric.** For ScanRefer (Chen et al., 2020), following previous work, we use Acc@$m$IoU as the evaluation metric, where $m \in \{0.25, 0.5\}$. This metric represents the ratio of the predicted bounding boxes whose Intersection over Union (IoU) with the ground-truth (GT) bounding boxes is larger than $m$. For Sr3D and Nr3D (Achlioptas et al., 2020), the ground truth bounding boxes are available, and the model only needs to identify the described object from all the bounding boxes. Therefore, the evaluation metric of these two datasets is accuracy, *i.e.*, the percentage of the correctly selected target object.

### 4.2 Quantitative Comparison

We compare our 3DRP-Net with other state-of-the-art methods on these three 3D visual grounding benchmarks.

**ScanRefer.** Table 1 shows the performance on ScanRefer. 3DRP-Net outperforms the best two-stage method by +4.20 at Acc@0.25 and +4.40 at Acc@0.5 and exceeds the best one-stage method by +2.45 at Acc@0.25 and +2.47 at Acc@0.5. Even when compared to 3DJCG, which utilizes an extra Scan2Cap (Chen et al., 2021) dataset to assist its training, our 3DRP-Net still shows superiority in all metrics. Specifically, for the "Multiple" subset, 3DRP-Net achieves +2.66 and +2.34 gains when compared with the advanced one-stage model in terms of Acc@0.25 and Acc@0.5, which validates the proposed 3DRP-MA module is powerful for modeling complex relative position relations in 3D space and significantly contributes to distinguishing the described target object from multiple interfering objects.

Table 1: Comparisons with state-of-the-art methods on *ScanRefer*. We highlight the best performance in **bold**.

| Methods | | Extra | Unique | | Multiple | | Overall | |
|---|---|---|---|---|---|---|---|---|
| | | | Acc@0.25 | Acc@0.5 | Acc@0.25 | Acc@0.5 | Acc@0.25 | Acc@0.5 |
| *Two-stage:* | ScanRefer | - | 67.64 | 46.19 | 32.06 | 21.26 | 38.97 | 26.10 |
| | TGNN | - | 68.61 | 56.80 | 29.84 | 23.18 | 37.37 | 29.70 |
| | InstanceRefer | - | 77.45 | 66.83 | 31.27 | 24.77 | 40.23 | 32.93 |
| | SAT | 2D assist | 73.21 | 50.83 | 37.64 | 25.16 | 44.54 | 30.14 |
| | 3DVG-Transformer | - | 77.16 | 58.47 | 38.38 | 28.70 | 45.90 | 34.47 |
| | MVT | - | 77.67 | 66.45 | 31.92 | 25.26 | 40.80 | 33.26 |
| | 3DJCG | Scan2Cap | 78.75 | 61.30 | 40.13 | 30.08 | 47.62 | 36.14 |
| | ViL3DRel | - | 81.58 | **68.62** | 40.30 | 30.71 | 47.94 | 37.73 |
| *One-stage:* | 3D-SPS | - | 81.63 | 64.77 | 39.48 | 29.61 | 47.65 | 36.43 |
| | **3DRP-Net (Ours)** | - | **83.13** | 67.74 | **42.14** | **31.95** | **50.10** | **38.90** |

Table 2: Comparisons with state-of-the-art methods on *Nr3D* and *Sr3D*. We highlight the best performance in **bold**.

| Method | Nr3D | | | | | Sr3D | | | | |
|---|---|---|---|---|---|---|---|---|---|---|
| | Easy | Hard | View Dep | View Indep | Overall | Easy | Hard | View Dep | View Indep | Overall |
| ReferIt3DNet | 43.6 | 27.9 | 32.5 | 37.1 | 35.6 | 44.7 | 31.5 | 39.2 | 40.8 | 40.8 |
| InstanceRefer | 46.0 | 31.8 | 34.5 | 41.9 | 38.8 | 51.1 | 40.5 | 45.4 | 48.1 | 48.0 |
| 3DVG-Transformer | 48.5 | 34.8 | 34.8 | 43.7 | 40.8 | 54.2 | 44.9 | 44.6 | 51.7 | 51.4 |
| LanguageRefer | 51.0 | 36.6 | 41.7 | 45.0 | 43.9 | 58.9 | 49.3 | 49.2 | 56.3 | 56.0 |
| SAT | 56.3 | 42.4 | 46.9 | 50.4 | 49.2 | 61.2 | 50.0 | 49.2 | 58.3 | 57.9 |
| 3D-SPS | 58.1 | 45.1 | 48.0 | 53.2 | 51.5 | 65.4 | 56.2 | 49.2 | 63.2 | 62.6 |
| MVT | 61.3 | 49.1 | 54.3 | 55.4 | 55.1 | 66.9 | 58.8 | 58.4 | 64.7 | 64.5 |
| ViL3DRel | 70.2 | 57.4 | 62.0 | 64.5 | 64.4 | 74.9 | 67.9 | 63.8 | 73.2 | 72.8 |
| **3DRP-Net(Ours)** | **71.4** | **59.7** | **64.2** | **65.2** | **65.9** | **75.6** | **69.5** | **65.5** | **74.9** | **74.1** |

**Nr3D/Sr3D.** Note that the task of Nr3D/Sr3D is different from ScanRefer, which aims to identify the described target object from all the given ground-truth bounding boxes. Therefore, the soft-labeling strategy and the keypoint sampling module are removed. We only verify the effectiveness of 3DRP-MA on these two datasets. Besides, the data augmentation methods in ViL3DRel (Chen et al., 2022) are also used in our training phase for a fair comparison. The accuracy of our method, together with other state-of-the-art methods, is reported in Table 2. 3DRP-Net achieves the overall accuracy of 65.9% and 74.1% on Nr3D and Sr3D, respectively, which outperforms all existing methods by a large margin. In the more challenging "Hard" subset, 3DRP-Net significantly improves the accuracy by +2.3% in Nr3D and +1.6% in Sr3D, again demonstrating our method is beneficial for distinguishing objects by capturing the relative spatial relations.

### 4.3 Ablation Study

We conduct ablation studies to investigate the contribution of each component. All the ablation study results are reported on the ScanRefer validation set.

**Relation Modeling Module.** We compared our proposed 3DRP-MA with the relation modules in other 3D visual grounding methods. For fair comparisons, we also introduce distances in x, y, z coordinates and Euclidean space to other relation modules. The results are provided in Table 3, comparing rows 1, 2 and 6, our 3DRP-MA is far superior to the relation modules in 3DVG-Trans and 3DJCG, and the performance improvement mainly comes from the subsets that rely on relative relationship reasoning for localization, namely the "One-Rel" and "Multi-Rel" subsets.

**Relative Position Encoding.** In Sec.3.2.3, we discuss the complexity of relative relations in 3D space and propose four relative position encodings based on relative distance in x,y,z coordinates ($D_{xyz}$), and the Euclidean metric ($D_e$), respectively. From Table 3, both $D_{xyz}$ and $D_e$ can bring significant improvement for subsets that require relative relation reasoning. Row 6 demonstrates that considering relative relations from multiple directions further helps capture comprehensive and sufficient object relations and distinguish the target object from multiple distractors.

Table 3: Ablation studies on relation position encoding and different relation modeling modules. None-Rel/One-Rel/Multi-Rel represent subsets that contain zero/one/multiple relation descriptions in the original Multiple set of ScanRefer, and the relative percentage improvements compared to the different settings are marked in green.

| Row | $D_e$ | $D_{xyz}$ | Rel Module | Overall | Multiple | None-Rel | One-Rel | Multi-Rel |
|-----|-------|-----------|------------|---------|----------|----------|---------|-----------|
| 1 | ✓ | ✓ | 3DVG-Transformer | 36.85 | 30.16 | 34.89(+2.95%) | 32.51(+5.51%) | 28.03(+6.60%) |
| 2 | ✓ | ✓ | 3DJCG | 36.43 | 29.62 | 35.51(+1.15%) | 31.87(+7.62%) | 27.35(+9.25%) |
| 3 | ✗ | ✗ | 3DRP-MA | 32.74 | 26.39 | 34.18(+5.09%) | 28.39(+20.82%) | 23.94(+24.81%) |
| 4 | ✓ | ✗ | 3DRP-MA | 36.43 | 30.26 | 35.47(+1.27%) | 32.54(+5.41%) | 28.10(+6.33%) |
| 5 | ✗ | ✓ | 3DRP-MA | 37.13 | 30.56 | 35.30(+1.76%) | 32.87(+4.35%) | 28.46(+4.99%) |
| 6 | ✓ | ✓ | 3DRP-MA | **38.90** | **31.91** | **35.92** | **34.30** | **29.88** |

Table 4: Ablation studies on 3DRP-MA in each transformer layer and pair-aware relation attention.

| Row | O1-R | R-O2 | $SA_1$ | $CA$ | $SA_2$ | Acc@0.25 | Acc@0.5 |
|-----|------|------|--------|------|--------|----------|---------|
| 1 | ✗ | ✓ | ✓ | ✓ | ✓ | 48.83 | 38.46 |
| 2 | ✓ | ✗ | ✓ | ✓ | ✓ | 48.30 | 37.56 |
| 3 | ✓ | ✓ | ✓ | ✗ | ✗ | 46.70 | 36.10 |
| 4 | ✓ | ✓ | ✓ | ✓ | ✗ | 48.72 | 37.59 |
| 5 | ✓ | ✓ | ✓ | ✓ | ✓ | **50.10** | **38.90** |

Table 5: Ablation studies on the labeling strategies.

| Strategy | $N_s$ | Acc@0.25 | Acc@0.5 |
|----------|-------|----------|---------|
| IoUs | Original | 48.20 | 38.06 |
|      | Linear | 48.82 | 37.50 |
| Hard | 1 | 47.36 | 37.25 |
|      | 4 | 47.29 | 37.68 |
|      | 8 | 47.30 | 37.26 |
| Soft | 12 | 49.13 | 38.46 |
|      | 24 | **50.10** | **38.90** |
|      | 36 | 49.64 | 38.55 |

**Pair-aware relation attention.** The typical description of a spatial relation can be expressed as *"Object 1-Relation-Object 2"*. Our pair-aware relation attention can be considered as the sum of two scores: *Object 1-to-Relation* (O1-R) and *Relation-to-Object 2* (R-O2). To further verify the superiority of capturing the relation in the context of an object pair, we ablate the two scores, and the results are illustrated in Table 4. From rows 1, 2 and 5, both O1-R and R-O2 terms benefit the 3D visual grounding task by capturing the relative relations, and the joint use of O1-R and R-O2 provides a more comprehensive understanding of spatial relation description and leads to the best performance.

**3DRP-MA in each layer.** We study the effect of each 3DRP-MA module in the transformer layer. $SA_1$, $CA$ and $SA_2$ respectively denote whether to replace the self-attention before interacting with seed points, the cross-attention for key points and seed points, and the self-attention before interacting with language. Row 3 to 5 in Table 4 add each 3DRP-MA in turns and the performance is gradually improved to 50.10% and 38.90%.

**Soft-labeling Strategy.** Table 5 presents the performance of different labeling strategies. In hard-labeling, $N_s$ represents the number of key points whose IoU is in the top $N_s$ and greater than 0.25, which are labeled as 1. In soft-labeling, $N_s$ is a hyperparameter in Eq.5, which controls the number of soft labels. To further demonstrate that our proposed strategy improves stability and discrimination, we also use IoUs score as a label. The "original" setting directly uses IoUs as a label, while the "linear" setting stretches IoUs linearly to the range of 0 to 1 to enhance discrimination. Compared to hard-labeling and IoUs methods, our soft-labeling strategy improves discrimination and stability. Using the "original" IoUs method lacks discrimination power and stability due to the unbalanced loss on different samples. Even using linear scaling to enhance discrimination power, this instability cannot be eliminated. Our method alleviates these problems with a discriminative constant distribution and shows comprehensive superiority in Table 5.

## 5 Conclusion

In this paper, we propose a relation-aware one-stage model for 3D visual grounding, referred to as 3D Relative Position-aware Network (3DRP-Net). 3DRP-Net contains novel 3DRP-MA modules to exploit complex 3D relative relations within point clouds. Besides, we devise a soft-labeling strategy to achieve disambiguation while promoting a stable and discriminative learning process. Comprehensive experiments reveal that our 3DRP-Net outperforms other methods.

# 6 Limitations

The datasets of 3D visual grounding task are all stem from the original ScanNet dataset which brings generalization to other scene types into question. More diverse benchmarks are important for the further development of the field of 3D visual grounding.

# Acknowledgments

This work was supported in part by National Natural Science Foundation of China under Grant No.62222211, Grant No.61836002 and Grant No.62072397.

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

# A   Qualitative Analysis

In this section, we provide some visualization results in ScanRefer (Chen et al., 2020) for qualitative analysis.

## A.1   Analysis on Success Cases

To better understand our 3DRP-Net, we visualize some success cases and comparisons with the other one-stage method (Luo et al., 2022) in Figure 4. From (a,b,c), both 3D-SPS (Luo et al., 2022) and our 3DRP-Net accurately locate the target object when the description does not involve too many relative position relations and there are not many interfering objects in the scene. However, as shown in (d,e,f), when the relative position relation between objects is necessary for distinguishing the target object from multiple objects of the same category, the previous one-stage method 3D-SPS is often confused by distractors. By modeling the relative position in 3D space, our 3DRP-Net is able to fully leverage the relative position descriptions in the sentence for reasoning, which bring more precise localization.

## A.2   Analysis on Failure Cases

To conduct a comprehensive qualitative evaluation, we further elaborate on the failure cases and discuss them in detail. These reasons for our 3DRP-Net prediction errors can be roughly summarized into three categories:

- **Ambiguous annotations.** Due to the complexity and irregularity of 3D scenes, ambiguous descriptions are difficult to be completely avoided in 3D visual grounding datasets. There may be multiple objects in a scene that match the description, but only one of them is considered correct by the annotation. As shown in the cases (1,2,3) of Figure 5, both ground-truth objects and our predicted objects semantically match the natural language descriptions, but according to the ground truth box annotations, our predictions are completely wrong.

- **Challenging target object.** In 3D point clouds, some objects are inherently difficult to identify because of obscured or missing surfaces. In case 4 of Figure 5, the described target object is a ***cabinet***, but the point cloud in the ground truth box is seriously missing,

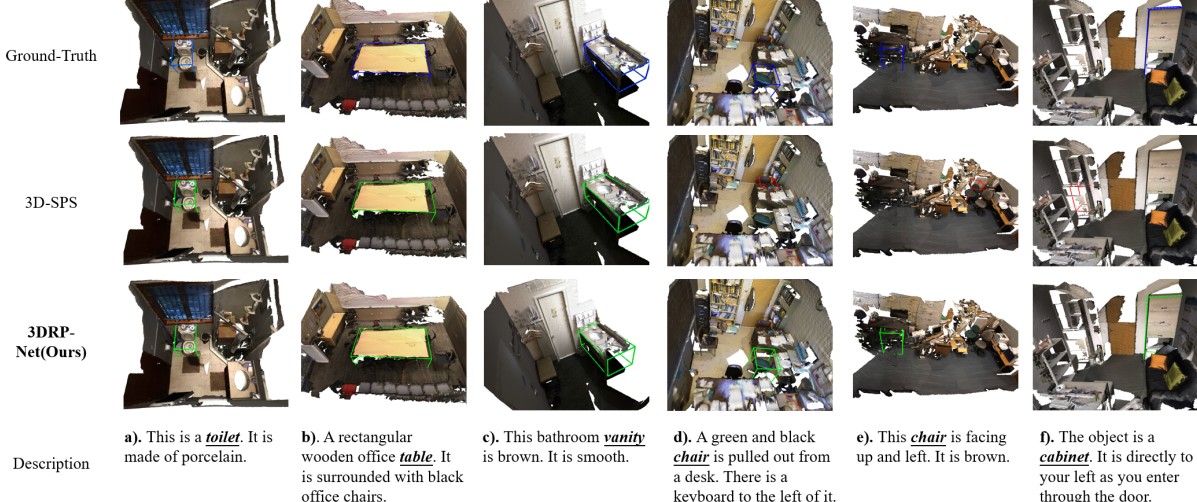

Ground-Truth

3D-SPS

3DRP-Net(Ours)

Description

**a).** This is a _toilet_. It is made of porcelain.

**b).** A rectangular wooden office _table_. It is surrounded with black office chairs.

**c).** This bathroom _vanity_ is brown. It is smooth.

**d).** A green and black _chair_ is pulled out from a desk. There is a keyboard to the left of it.

**e).** This _chair_ is facing up and left. It is brown.

**f).** The object is a _cabinet_. It is directly to your left as you enter through the door.

Figure 4: The visualization results of some success cases. The blue/green/red colors indicate the ground truth/correct/incorrect boxes.

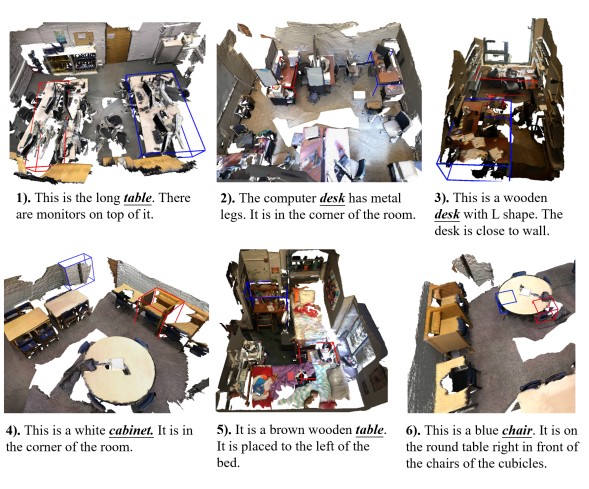

**1).** This is the long _table_. There are monitors on top of it.

**2).** The computer _desk_ has metal legs. It is in the corner of the room.

**3).** This is a wooden _desk_ with L shape. The desk is close to wall.

**4).** This is a white _cabinet_. It is in the corner of the room.

**5).** It is a brown wooden _table_. It is placed to the left of the bed.

**6).** This is a blue _chair_. It is on the round table right in front of the chairs of the cubicles.

Figure 5: The visualization results of some failure cases. The ground-truth boxes are labeled in blue and the incorrectly predicted boxes are marked in red.

which makes it very difficult to identify the _cabinet_ in the scene.

- **Challenging auxiliary objects.** 3D visual grounding task often requires the relations between the target object and auxiliary objects to assist the localization. The challenging auxiliary objects may result in an incorrect prediction. As shown in case 5 of Figure 5, the target _table_ is on "the left of the bed", but the left and right side of a bed are difficult to distinguish, which requires identifying the direction of the bed according to the position of pillows. This reasoning process is too complex for our model, and our prediction actually found the table on the right side of a bed. In case 6, the auxiliary object is "chair of the cubicles", which is challenging for the model to recognize.

## B    Statistics of Relative Position Words

To further illustrate that relative position relation is a general and fundamental issue in 3D visual grounding task, we count some common words representing relative spatial relations in three 3D visual grounding datasets (i.e., ScanRefer (Chen et al., 2020), Nr3D (Achlioptas et al., 2020) and Sr3D (Achlioptas et al., 2020)) in Figure 6 and 7. From Figure 6, in ScanRefer, at least 97% descriptions contain relative position relations, and more than 63% sentences use multiple relative position relations to indicate the target object. Besides, about 90% sentences utilize the relative position words in Nr3D, and almost all the samples in Sr3D require relative position relations between objects for localization. As shown in Figure 7, in ScanRefer and Nr3D, which collected human utterances as descriptions, most of the commonly used relative position words appear in the sentences. This further demonstrates the importance of modeling relative position relations from different perspectives.

## C    Implementation Details.

We adopt the pre-trained PointNet++ (Qi et al., 2017) and the language encoder in CLIP (Radford et al., 2021) to extract the features from point clouds and language descriptions, respectively, while the rest of the network is trained from

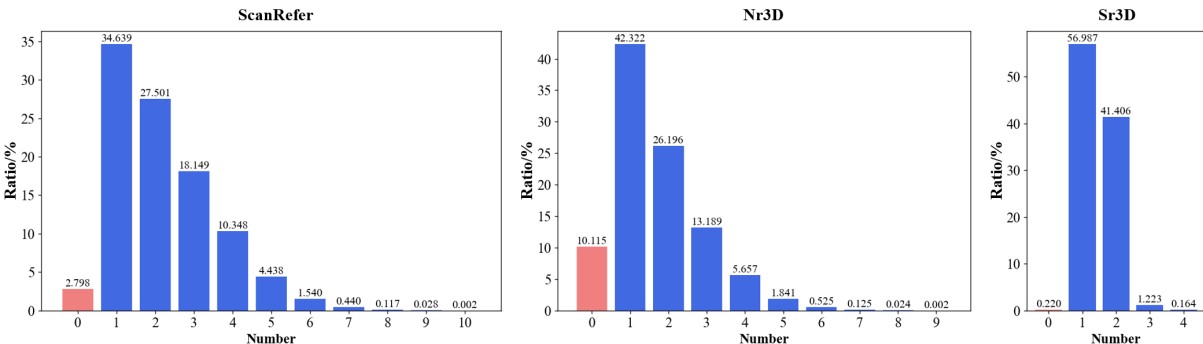

Figure 6: Ratio of sentences containing the specific number of relative position words in three 3D visual grounding datasets.

scratch. We set the dimension $d$ in all transformer layers to 384. The layer number of the transformer is set to 4. Our model is trained in an end-to-end manner with the AdamW (Kingma and Ba, 2014) optimizer and a batch size of 15 for 36 epochs. The initial learning rates of all transformer layers and the rest of the model are set to $1e - 4$ and $1e - 3$, and we use the cosine learning rate decay strategy to schedule the learning rates. The seed point number $M$ and keypoint number $M_0$ are set to 1024 and 256. For the soft-labeling strategy, the label number $N_s$ is assigned as 24. In the piecewise index function, we set the $\alpha : \beta : \gamma = 1 : 2 : 4$, and the $\beta$ is assigned as 20. When calculating the relative position index, the coordinates of all points are linearly scaled to $[0, 100]$.

In the ablation study, we further divided the "Multiple" set of ScanRefer into "Non-Rel/One-Rel/Multi-Rel" subsets according to the number of relational descriptions in the sentences. Specifically, we follow the statistical method in Sec. B to count some common words representing relative spatial relations.

## D Prior Methods for Comparison

In order to validate the effectiveness of the proposed 3DRP-Net, Sec. 4.2 comprehensively compare it to many previous state-of-the-art methods: 1) ReferIt3DNet (Achlioptas et al., 2020) 2) ScanRefer (Chen et al., 2020); 3) TGNN (Huang et al., 2021); 4) InstanceRefer (Yuan et al., 2021); 5) LanguageRefer (Roh et al., 2022); 6) SAT (Yang et al., 2021); 7) 3DVG-Trans (Zhao et al., 2021b); 8) MVT (Huang et al., 2022); 9) 3D-SPS (Luo et al., 2022); 10) 3DJCG (Cai et al., 2022); 11) ViL3DRel (Chen et al., 2022)

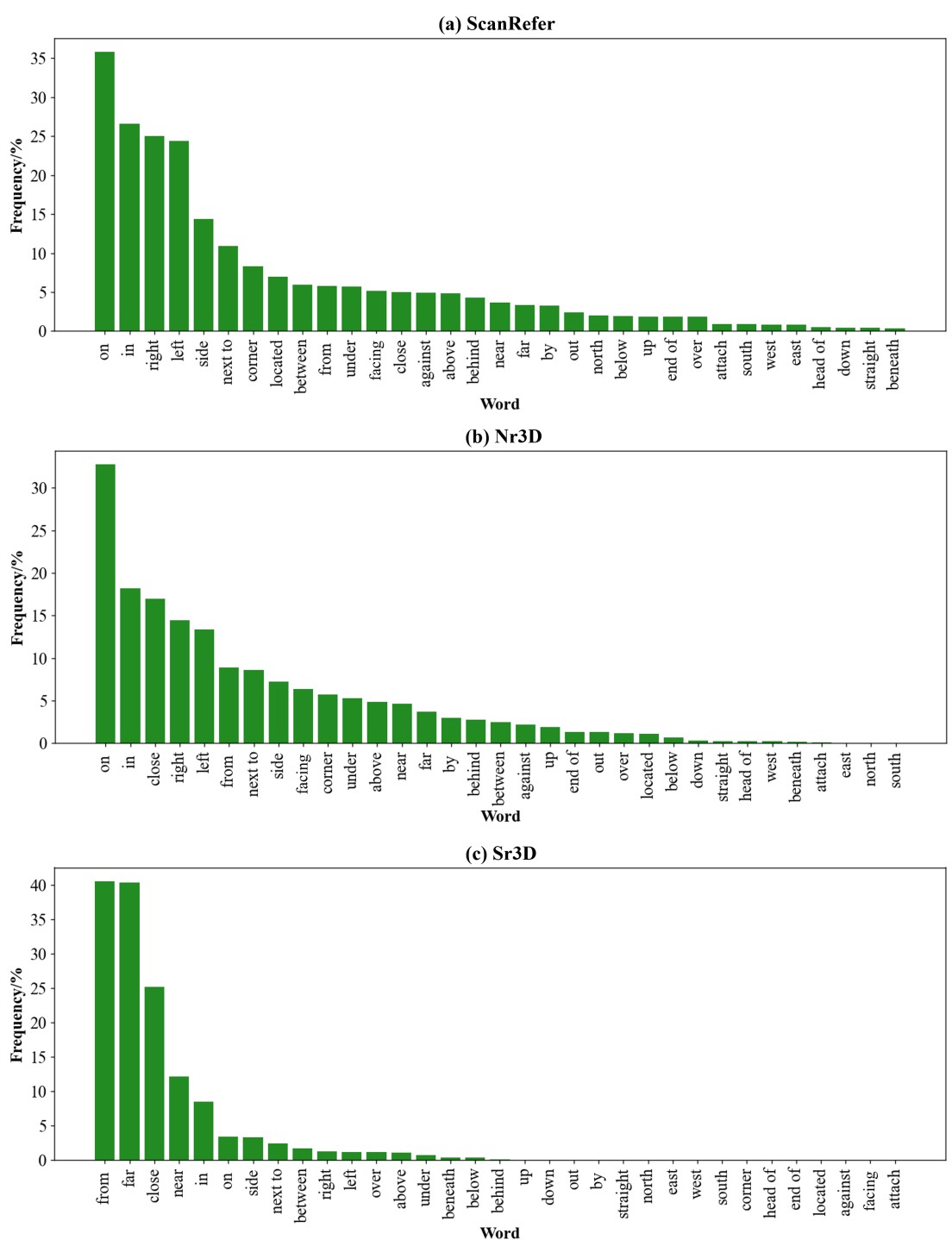

Figure 7: Frequency of some commonly used relative position words in three 3D visual grounding datasets.