# OpenReview forum: "3DRP-Net: 3D Relative Position-aware Network for 3D Visual Grounding"
_EMNLP/2023/Conference — EMNLP 2023 Main_

### Official Review · Reviewer_bfbG · 2023-07-31

**Typos Grammar Style And Presentation Improvements:** 1. Fig.3 can be more clear if supplem…
**Soundness:** 3

**Excitement:**

4: Strong: This paper deepens the understanding of some phenomenon or lowers the barriers to an existing research direction.

**Paper Topic And Main Contributions:**

This work introduces the new model 3DRP-Net, aimed at improving 3D visual object grounding given point cloud of the scene and natural language input.

The proposed model uses a relatively novel one-stage structure to avoid unstable and time-consuming object proposals generation stage while employing relative position encoding instead of absolute position encoding.

Evaluation is run on the benchmark datasets ScanRefer and parts of ReferIt3D. Automatic evaluations show model significantly outperforming SOTA methods.

The method enhances spatial reasoning abilities of the 3D-SPS model, making it better suited to reference resolution and 3d spatial relation reasoning.

**Questions For The Authors:**

A. Is there a better way to capture object spatial relations than the proposed pairwise mechanism? Perhaps accounting for multi-to-multi relationships.

**Reasons To Accept:**

1. The paper is generally well-written, the motivation is clear and straightforward.

2. The method proves to be substantially better than existing baselines though thorough evaluations on ScanRefer and Nr3D/Sr3D datasets.

3. Model offers structural simplicity compared to existing ones (though this is largely inherited from the previous 3D-SPS model with minor modifications.

**Reasons To Reject:**

1. All evaluation are run on indoor scenes of everyday environments in a western setting. It would be interesting to see whether the model can adapt to different scenarios/out-of-distribution cases.

**Reproducibility:**

4: Could mostly reproduce the results, but there may be some variation because of sample variance or minor variations in their interpretation of the protocol or method.

**Reviewer Confidence:**

3: Pretty sure, but there's a chance I missed something. Although I have a good feel for this area in general, I did not carefully check the paper's details, e.g., the math, experimental design, or novelty.

---

> ### Author Rebuttal · Authors · 2023-08-28
>
> **Q1: “All evaluations are run on indoor scenes of everyday environments in a western setting”**
>
> Existing visual grounding datasets are all constructed based on indoor scenes. Our work focuses on proposing a 3D visual grounding architecture with stronger relative relation modeling capabilities and compares it with previous visual grounding methods. Extending visual grounding to more scenarios will be a very important development direction in the 3D-language field. Once more datasets of other scenarios are released, we will evaluate the applicability of previous methods and ours in more scenarios.
>
> **Q2: “Is there a better way to capture object spatial relations than the proposed pairwise mechanism?”**
>
> Modeling the relationship between objects pair multiple times via the stacked L-layer transformer layers can implicitly model the relationship among multiple objects. More explicit multi-to-multi relation modeling will be an interesting direction for future work. Considering that we are the first to explore relative position modeling under the plain one-stage framework, we believe our method is inspiring and would be a good starting point for future works.

---

### Official Review · Reviewer_Qbww · 2023-07-31

**Soundness:** 3

**Excitement:**

3: Ambivalent: It has merits (e.g., it reports state-of-the-art results, the idea is nice), but there are key weaknesses (e.g., it describes incremental work), and it can significantly benefit from another round of revision. However, I won't object to accepting it if my co-reviewers champion it.

**Paper Topic And Main Contributions:**

The paper proposes a 3D relative position-aware network (3DRP-Net) for 3d object grounding given language sentences. 3DRP-Net is a one-stage model with two new components compared to existing one-stage methods: i) a 3D relative position multi-head attention to enhance the understanding of 3d pairwise object relation; ii) a soft-labeling strategy to improve training with weighted losses for predicted object proposals. 3DRP-Net achieved state-of-the-art performance on three benchmarks.

**Questions For The Authors:**

See reasons to reject.

**Reasons To Accept:**

- The paper is well-written and easy to follow.

- The 3d relative position multi-head attention provides a simple but effective way to inject location information into transformers.

- The proposed model outperforms existing methods, and the ablations demonstrate the effectiveness of the new components.

**Reasons To Reject:**

A. The 3DRP-Net is built upon the one-stage model 3D-SPS and improves it with the proposed relation-aware attention and soft-labeling training. However, comparing results in the ablations with 3D-SPS shows that the two components do not work or have limited improvement.

Table 2: The main difference between the model in row 3 and 3D-SPS should be the training strategy. However, the result in row 3 is much worse than 3D-SPS, indicating the proposed training strategy does not work.

Table 5: Since 3D-SPS used a hard-labeling training strategy, the main difference between the models in Table 5 Hard block and the 3D-SPS is the proposed relation-aware attention. However, the results in Table 5 are slightly worse than 3D-SPS on Acc@0.25, indicating that the proposed relation-aware attention is not that useful.

B. The introduction and related work sections do not discuss existing works sufficiently though some of them are compared in the experimental tables, such as SAT and ViL3DRel.

In particular, ViL3DRel is one of the most similar works that also uses 3d relative spatial relations in transformer attention layers, which should be discussed and compared in depth.
It is unclear whether the approach in ViL3DRel could work in the one-stage framework and how much gains the proposed method can outperform ViL3DRel.


C. The method contains some major limitations that are not discussed or better designs are required.

1) The proposed method only captures pairwise spatial relations between objects, however, there are triplet relations in the sentences such as “object 1 is between object 2 and object 3”.

2) It is unclear why the authors use learnable embedding to encode the relative distances, which can lose fine-grained information. For example, it may fail if the goal is “find the closest chair to the door” and the two chairs are in 2.0 and 2.49 distances to the door, since the round operation would make the two distance embeddings the same.

3) The D_{xyz} might not correspond to relative directions such as left, right etc. since the object orientations are unknown. The relative directions should be relevant to the object orientations.

**Reproducibility:**

4: Could mostly reproduce the results, but there may be some variation because of sample variance or minor variations in their interpretation of the protocol or method.

**Reviewer Confidence:**

5: Positive that my evaluation is correct. I read the paper very carefully and I am very familiar with related work.

**Typos Grammar Style And Presentation Improvements:**

- The presentation of table 3 can be improved. It took me quite a while to understand the gains marked in green in row 1-5, which denote the relative improvements of row 6 over each row. It would be easier to understand if the relative performance decreases are presented for each row.

- Grammar mistake: L322 in Sec 3.3 "Due to" -> "Because".

---

> ### Author Rebuttal · Authors · 2023-08-28
>
> **Q1.1: “The 3DRP-Net is built upon the one-stage model 3D-SPS. Comparing results in the ablations with 3D-SPS shows that the two components do not work or have limited improvement”**
>
> Our approach does not simply improve the modules and training strategies under the structure of 3D-SPS, and the main architecture of our model is also different from 3D-SPS.
>
> The difference in architecture between Ours and 3D-SPS: 3D-SPS introduces a complex structure (The TPM module in Figure 4 of 3D-SPS paper) which encodes 3D-vision and language modalities via two parallel branches and devises sophisticated modality interaction, along with a progressive key point selection strategy to filter out target objects. In contrast, our approach only sequentially stacks two vanilla transformer decoders to integrate scene and language information into key-point features. Therefore, the ablation results of our approach are not comparable to the results of 3D-SPS.
>
> Why design such an architecture: Our experiments found that the progressive key-point selection strategy in 3D-SPS limits the comprehensive modeling of relative positions between objects. The text information in 3D-SPS interacting with the vision feature will disturb the consistency of the textual condition of the object relation modeling in different layers. The sophisticated architecture of 3D-SPS is not suitable for exploring more effective relation modeling modules. Our further experiments show that simply replacing the attention in 3D-SPS with 3DRP-MA, the Acc@0.25, and Acc@0.5 under ScanRefer would drop from 47.65, 36.43 to 39.89, 30.01.
>
> In summary, our contribution is not only a new relative relationship modeling module and soft label strategy for the one-stage model but also a plainer and more adaptable one-stage architecture for 3D visual grounding.
>
> **Q1.2: Comparisons between Table 5 and Row 3 in “Table 2” with 3D-SPS**
>
> Considering that Table 2 in our paper is the performance comparison on Nr3D and Sr3D, we suppose there is a typo. You may mean Row 3 in Table 3.
>
> As we discussed in Q1.1, in addition to the relative relationship module and soft label strategy, the network architecture of our method is different from 3D-SPS. Besides, the progressive key-point strategy in 3D-SPS is not used in our method, in order to capture the relations between objects more comprehensively.
>
> **Q2: “Related work sections do not discuss existing works sufficiently.” “ViL3DRel should be discussed and compared in depth”**
>
> Thanks for your suggestion, we will add more discussion in the related work of the future version.
>
> For ViL3DRel, its relationship modeling (the “Spatial Attention Matrix” in their paper) does not consider object features. Therefore, its spatial matrix would highlight all object pairs that have a similar relative relation according to the language inputs. However, our method realizes the interaction among object-relation-language and can pay more attention to the exact object relation mentioned in the sentence.
>
> Moreover, ViL3DRel adopts a continuous relative position encoding, which results in a computational complexity of $O(n^2 d)$, where $n$ is the number of tokens and $d$ is the dimension of the token. In the two-stage approach, each object is used as a token, so the $n$ is relatively small. In the one-stage method, many points are sampled as tokens, the $n$ will become very large. Therefore, adopting the relative position module of ViL3DRel in a one-stage method will also introduce a lot of additional computational complexity.
>
> **Q3.1: triplet relations in the sentences such as “object 1 is between object 2 and object 3”**
>
> This work mainly explores a plain model structure with strong relative relation modeling ability and a novel effective training strategy. Modeling the relationship between object pairs multiple times via the stacked L-layer transformer layer can implicitly model the relationship among multiple objects. More explicit relation modeling among more objects will be an interesting direction for future work. Considering we are the first to explore relative position modeling under the plain one-stage framework, we believe the current method is inspiring and would be a good starting point for future works.
>
> **Q3.2.1: “Why the authors use learnable embedding to encode the relative distances”**
>
> The discretized relative position embedding is a more computationally efficient method. Considering $n$ tokens with $d$ dimension, using discretized position representations can reduce the complexity of calculating the Relation Attention Matrix from $O(n^2 d)$ to $O(nkd)$. Since the one-stage method uses each point as a token, the $n$ is much larger than $k$. Our experiments show using discretized position embedding can effectively reduce computational complexity with no performance loss. Using continuous position encoding takes 50 hours in training on a single 3090 and the Acc@0.25 and Acc@0.5 on ScanRefer is 50.07, 38.91, while using the discretized relative position embedding, the training time reduces to 30 hours with 50.10, 38.90 accuracies.
>
> **Q3.2.2: “The round operation would make the two distance embeddings the same.”**
>
> In our implementation, we linearly scaled the coordinates of all points to [0:100]. The difference between 2.00 and 2.49 in the 3D space is also indistinguishable for humans, and the round operation is fine-grained enough for understanding relative relations in 3D scenes.
>
> In addition, the coordinates of points can be linearly scaled to a larger scale, which brings finer-grained relative distance embeddings, enabling more detailed relationship modeling in larger scenes. At the same time, by adjusting the hyperparameters in Equation 4, we can control the balance between computational complexity and granularity of position embeddings.
>
> **Q3.3 “$D_{xyz}$ might not correspond to relative directions such as left, right, etc.”**
>
> Thanks for your suggestion, the current statement is indeed vague and ambiguous. $D_{xy}$ represents the relative distance on the horizontal plane, and $D_z$ represents the vertical relationship. Since $D_x$ and $D_y$ embedding would interact with object features (which include object orientation information), the model can learn the correspondence between x-y coordinates and the left, right direction of the object. We will clarify this in the future version.
>
> **Typos:**
>
> Thanks for your suggestion, we will add more descriptions about the notation of relative improvements in the title of Table 3. The grammar mistake in L322 in Sec 3.3 will also be revised.

---

### Official Review · Reviewer_fZVS · 2023-08-05

**Paper Topic And Main Contributions:** 3D visual grounding is the problem to…
**Soundness:** 3

**Excitement:**

4: Strong: This paper deepens the understanding of some phenomenon or lowers the barriers to an existing research direction.

**Questions For The Authors:**

The dataset is not big. Are there other datasets you can also try out your method?

Can you scale up your method to several orders of more data?

== post rebuttal

The authors provided their explanations on the difficulties of addressing these questions. However, there is no new results or insights. I maintain my original rating.

**Reasons To Accept:**

The paper has several notable contributions, which constitute compelling reasons for its acceptance:

It introduces an innovative single-stage 3D visual grounding model, termed 3D Relative Position-aware Network (3DRP-Net). This model incorporates relative position relationships in the context of object pairs, thus enhancing the capability for spatial relation reasoning.

It has also developed a unique 3D Relative Position Multi-head Attention (3DRP-MA) module. This mechanism allows simultaneous modeling of spatial relations from multiple orientations in 3D space.

The paper also proposes a soft-labeling strategy to mitigate the ambiguity typically associated with identifying optimal key points in 3D objects.

Finally, the paper validates the effectiveness of the proposed model through a series of comprehensive experiments. The 3DRP-Net model attains state-of-the-art performance on three widely-used benchmark datasets: ScanRefer, Nr3D, and Sr3D.

**Reasons To Reject:**

While the paper did a good job taking a traditional approach -- developing a specialized model with ground truth data. A better approach is to develop a foundation model without relying on a large amount of labeled data. For example, the recent work on PaLI-X, RT-2, Kosmos-2 are all along this direction.

PaLI-X: On Scaling up a Multilingual Vision and Language Model
https://arxiv.org/abs/2305.18565

RT-2: Vision-Language-Action Models Transfer Web Knowledge to Robotic Control
https://arxiv.org/abs/2307.15818

Kosmos-2: Grounding Multimodal Large Language Models to the World
https://arxiv.org/abs/2306.14824

**Reproducibility:**

4: Could mostly reproduce the results, but there may be some variation because of sample variance or minor variations in their interpretation of the protocol or method.

**Reviewer Confidence:**

4: Quite sure. I tried to check the important points carefully. It's unlikely, though conceivable, that I missed something that should affect my ratings.

---

> ### Author Rebuttal · Authors · 2023-08-28
>
> **Q1： Developing a foundation model like PaLI-X, RT-2 and Kosmos-2**
>
> From the task-setting perspective, these methods (PaLI-X, PT-2, Kosmos-2) are primarily based on 2D vision. However, the real world is 3-dimensional, and human perception of the world also encompasses depth information. Compared to 2D vision, the language-3D vision we are focusing on has unique and crucial advantages, including more spatial information, comprehensive object shape information, and precise localization within the real 3D world. Moreover, 3D vision also presents its own challenges, such as the sparse and irregular nature of 3D point clouds and the more complex spatial relationships understanding and reasoning.
>
> From the model development perspective, these 2D vision-language methods (PaLI-X, PT-2, Kosmos-2) benefit from well-studied image model architectures and strong pre-trained 2D-vision models of different downstream tasks (such as detection, question answering, and grounding). However, in the realm of 3D vision, larger-scale datasets and pre-trained models are still being explored. Considering the unique advantages and challenges of 3D vision, as well as its wild applications and vast potential, we believe the exploration of plain 3D visual grounding architectures with stronger relative relation modeling capability is meaningful and promising.
>
> **Q2: “The dataset is not big. Can you scale up your method to several orders of more data?”**
>
> Currently, the larger-scale datasets for 3D-language tasks are very scarce. ScanRefer, Nr3D, and Sr3D are the biggest and most used datasets for 3D visual grounding. Other advanced methods for 3D visual grounding (all compared methods in our paper) also rely on these datasets.
>
> We firmly believe that larger datasets and more powerful pre-trained models are crucial for the development of 3D-language filed. This will be an important direction for future works.

---

### Meta-Review · Area_Chair_2bv5 · 2023-09-19

**Recommendation:** 5

**Metareview:**

This paper proposes a new model for referring expression grounding in 3D point clouds. Their model uses attention to model spatial relationships between objects, and relate them to the description.

Most reviewers agree that there are multiple noteworthy and exciting model contributions that result in compelling performance.

Most reviewers agree that the experiments used to evaluate the model are comprehensive and sufficient, and the work is overall sound.

---

### Decision · Program_Chairs · 2023-10-07

**Decision:**

Accept-Main

**Comment:**

This paper proposes a new model for referring expression grounding in 3D point clouds. Their model uses attention to model spatial relationships between objects, and relate them to the description.

Most reviewers agree that there are multiple noteworthy and exciting model contributions that result in compelling performance.

Most reviewers agree that the experiments used to evaluate the model are comprehensive and sufficient, and the work is overall sound.